# Testosterone and Bone Health in Men: A Narrative Review

**DOI:** 10.3390/jcm10030530

**Published:** 2021-02-02

**Authors:** Kazuyoshi Shigehara, Kouji Izumi, Yoshifumi Kadono, Atsushi Mizokami

**Affiliations:** Department of Integrative Cancer Therapy and Urology, Kanazawa University Graduate School of Medical Science, 13-1, Kanazawa, Ishikawa 920-8641, Japan; azuizu2003@yahoo.co.jp (K.I.); yskadono@yahoo.co.jp (Y.K.); mizokami@staff.kanazawa-u.ac.jp (A.M.)

**Keywords:** testosterone, men, osteoporosis, bone mineral density

## Abstract

Bone fracture due to osteoporosis is an important issue in decreasing the quality of life for elderly men in the current aging society. Thus, osteoporosis and bone fracture prevention is a clinical concern for many clinicians. Moreover, testosterone has an important role in maintaining bone mineral density (BMD) among men. Some testosterone molecular mechanisms on bone metabolism have been currently established by many experimental data. Concurrent with a decrease in testosterone with age, various clinical symptoms and signs associated with testosterone decline, including decreased BMD, are known to occur in elderly men. However, the relationship between testosterone levels and osteoporosis development has been conflicting in human epidemiological studies. Thus, testosterone replacement therapy (TRT) is a useful tool for managing clinical symptoms caused by hypogonadism. Many recent studies support the benefit of TRT on BMD, especially in hypogonadal men with osteopenia and osteoporosis, although a few studies failed to demonstrate its effects. However, no evidence supporting the hypothesis that TRT can prevent the incidence of bone fracture exists. Currently, TRT should be considered as one of the treatment options to improve hypogonadal symptoms and BMD simultaneously in symptomatic hypogonadal men with osteopenia.

## 1. Introduction

Serum testosterone levels decrease by 1% annually with age in elderly men [1], which may induce various clinical symptoms of late-onset hypogonadism (LOH) syndrome [2]. LOH syndrome is involved in a cluster of clinical symptoms, including depression, irritability, sexual dysfunction, decreased muscle mass and strength, and decreased bone mineral density (BMD), visceral obesity, and metabolic syndrome, which have been thought to be associated with aging [3,4]. These symptoms and signs often impair the quality of life (QOL) in elderly men and are considered a serious public health concern in the current aging society. Thus, testosterone replacement therapy (TRT) is expected to be one of the tools for improving these clinical conditions and QOL in men with LOH syndrome. Consequently, its clinical use has substantially increased over the past years [5].

In particular, osteoporosis often causes compression spine fractures and femoral neck fractures in elderly men, resulting in a decrease of activities of daily living (ADL) and QOL. Estrogen, which is important for maintaining BMD, decreases immediately in women during menopause. However, testosterone, which decreases slowly with age, plays an important role in maintaining BMD in men. Therefore, osteoporosis occurs more commonly in elderly women than in men [6,7]. The prevalence of osteoporosis increases with age at <10%, 13%, 18%, and 21% for 40, 70–75, 75–80, and >80 year old men in Japan, respectively [8]. Moreover, about 12 million people are estimated to suffer from osteoporosis [6,7]. The incidence frequency of femoral neck fracture is fourfold more in men than in women with osteoporosis [9]. Thus, osteoporosis prevention is an important issue in maintaining ADL and QOL in elderly men.

BMD has a close correlation with serum testosterone levels in men. Moreover, testosterone levels immediately decrease because of androgen deprivation therapy (ADT) for prostate cancer, resulting in a decrease of BMD and osteoporosis. In addition, estradiol (E2) converted from testosterone by aromatase is deeply related to BMD maintenance. A relative decrease in estrogen level due to ADT also poses a risk for BMD loss [10,11]. In general, BMD decreases by about 2%–8% in 1 year after the commencement of ADT [12]. Furthermore, ADT increases the risk of decreased BMD at five- to tenfold compared to prostate cancer patients with normal testosterone levels. A meta-analysis demonstrated that 9%–53% of osteoporosis incidence was caused by ADT [13]. Consequently, patients with ADT have a definite higher risk of sustaining a fracture. Furthermore, ADT can increase the risk of proximal femur fractures by 1.5- to 1.8-fold [14,15]. The BMD decrease in these patients is caused by a decline in serum testosterone and estrogen levels by ADT.

As aforementioned, the association between testosterone deficiency and BMD loss has been currently clarified. It is believed that TRT can contribute to maintaining and increasing BMD among hypogonadal men. However, the efficacy of TRT for bone health in hypogonadal men has been currently less in consensus and more conflicting [16,17,18,19,20,21]. Therefore, this article reviewed the relationship between testosterone and BMD in men and mentioned the benefits of TRT on BMD among hypogonadal men.

## 2. Materials and Methods

A review of PubMed, MEDLINE, and EMBASE databases was conducted to search for original articles, systematic review, and meta-analysis under key words as following; “testosterone” or “hypogonadism”, “bone mineral density” or “osteoporosis” or “bone fracture”, and “men”. There was no limitation on language, publication status, and study design. Papers published from January 1990 through to October 2020 were collected. We also checked the references of systematic reviews and meta-analyses carefully to identify additional original articles for inclusion. Two reviewers screened the search results, and the data were collected on 4 November 2020.

All papers suitable for three topics including “The Relationship between Hypogonadism and BMD in human”, “The Relationship between Testosterone and Bone Fracture”, and “The Effects of TRT on Bone Health among Hypogonadal Men” from the journal databases were adopted for the present analysis. 

## 3. Molecular Roles of Sex Hormones on Bone Metabolism

Testosterone is converted to highly active dihydrotestosterone (DHT) by 5α-reductase in the cytoplasm of target cells [22,23]. Consequently, DHT can induce androgenic activity by binding to the androgen receptor (AR). Moreover, testosterone is converted to E2 by aromatase. E2 binds to the estrogen receptor (ER) and exerts estrogenic action. ERα and ERβ are the two ER subtypes. ERα is mainly associated with bone metabolism [10,24].

AR is present in chondrocytes and osteoblasts, although its expression level widely varies by age and bone sites. Testosterone acts directly on osteoblasts by AR and can consequently promote bone formation [17]. In addition, testosterone has some indirect effects on bone metabolism through various cytokines and growth factors [17,25,26,27,28] (Figure 1). Furthermore, testosterone can increase AR expression level in osteoblasts, resulting in differentiation promotion and osteoblast and chondrocyte apoptosis proliferation [17,29]. Consequently, less evidence supporting the hypothesis that testosterone has any direct effects on osteoclasts has been shown [30]. 

In addition, testosterone deficiency promotes the activation of nuclear factor kappa-B ligand (RANKL) production from osteoblasts, which contributes to the promotion of the differentiations and functions in osteoclasts. Increased RANKL level progresses bone resorption and decreases BMD [25,26]. Thus, testosterone positively regulates the expression of insulin-like growth factor-1 (IGF-1) and IGF-binding protein (IGF-BP) in osteoblasts. The differentiation and proliferation of chondrocytes and osteoblasts are induced by IGF and IGF-BP, and the suppression of apoptosis of chondrocytes promotes bone formation. Moreover, testosterone activates the expression of transforming growth factor-β (TGF-β) in osteoblasts and promotes the differentiation of osteoblasts [27]. Testosterone suppresses the activity of interleukins (IL)-6, which activates osteoclasts and promotes bone resorption. However, testosterone deficiency decreases in BMD through increased IL-6 activation [28]. 

E2 and ERα also play important roles in maintaining BMD in men and women. Estrogen has a greater effect than androgen in inhibiting bone resorption in men. Consequently, men with loss of ERα function exhibit extremely low BMD [31]. Male patients with aromatase deficiency have a marked decrease in BMD in trabecular and cortical bone. Thus, estrogen replacement therapy in these patients can improve BMD [32,33]. E2 generally regulates apoptosis and function of osteoclast, which contributes to BMD maintenance. Moreover, E2 deficiency may accelerate osteoclast apoptosis by increased tumor growth factor-β production [34,35]. IL-1, IL-6, IL-7, IGF-1, nuclear factor-κB (NF-κB), RANKL, and tumor necrotic factor-α (TNFα) are the E2 target genes [36,37,38,39]. However, E2 deficiency increases IL-6, which reduces osteoblast proliferation and activity while increasing osteoclastic activity and increasing the expression of RANKL-mediated osteoclastogenesis [36,37]. Some experimental data showed that estrogen decrease also induces inflammatory cytokine, IL-1, and TNFα, resulting in BMD loss. However, this phenomenon does not occur in IL-1 receptor- or TNFα-deficient mice [38,39].

## 4. The Relationship between Hypogonadism and BMD in Humans

The apparent relationship between testosterone deficiency and low BMD has been currently established [5]. In particular, this relationship is much stronger in young adult men with moderate to severe hypogonadism [40,41]. However, epidemiological information on male osteoporosis attributed to hypogonadism, especially in elderly men, has been less available. Therefore, the prevalence of hypogonadism among men with osteoporosis or bone fracture has not been clarified.

Some previous studies demonstrated that hypogonadism was the cause of male osteoporosis (6.9%–58%) [42,43,44,45,46,47] (Table 1). However, these studies are limited by their small sizes and various potential biases. In addition, case–control studies comparing the prevalence of hypogonadism between subjects with osteoporosis and controls have been limited. 

A case–control study found that the prevalence rates of hypogonadism in men with osteoporotic fracture and control were 58% and 18%, respectively [42]. A cross-sectional and longitudinal study involving 2447 community-dwelling elderly men stated that the prevalence rates of hypogonadism among men with osteoporosis and normal BMD were 6.9% and 3.2%. Conversely, the prevalence of osteoporosis in men with hypogonadism was significantly higher in those with eugonadism (12.3% and 6.0%, respectively) [45]. Furthermore, a recent meta-analysis involving 300 patients from five case–control studies revealed that no significant difference was observed in testosterone level in both cases and controls [48]. Fewer exercise habits, cigarette smoking, various medications, and underlying diseases (e.g., metabolic syndrome, especially in elderly men) may modify the exact relationship between testosterone and osteoporosis, resulting in conflicting data. Further case–control studies involving a large number of subjects are likely required to better clarify whether prevalence of osteoporosis is higher in hypogonadal compared with eugonadal men. 

In addition, whether low testosterone is a potential risk factor for developing male osteoporosis in men is still unclear. Several large-population observational trials have been performed to investigate the effects of testosterone deficiency on osteoporosis risk factors in men [45,49,50,51,52,53,54,55,56,57,58] (Table 2). Some previous studies supported the potential relationship between testosterone decline and low BMD [45,49,51,54], and other studies denied this association [50,52,53,55,56,57,58]. Conversely, some previous studies found that E2 was significantly correlated with BMD loss in elderly men [45,49,50,54,55,56,57], supporting the hypothesis that E2 is also significantly correlated with BMD in men. In men, 85% of serum E2 derives from testosterone conversion by aromatase [59]. Therefore, low testosterone leads to low E2 production via aromatase, which could provoke deleterious effects on BMD, but also on bone quality as assessed in vivo by some different diagnostic tools [60,61]. A previous study demonstrated that 12-month administration of aromatase inhibitor in elderly men with hypogonadism resulted in a further decrease in BMD [62].

The conflicting results on the role of testosterone in BMD may also be because serum testosterone levels do not always reflect local testosterone levels within bone tissues and localized testosterone metabolism. A sub-analysis of the prospective MrOS study in Sweden involving 631 elderly men showed that glucuronidated androgen metabolites, but not serum TT, had a significant correlation with BMD in elderly men, suggesting that localized testosterone levels may have an important role in maintaining BMD [63].

## 5. The Relationship between Testosterone and Bone Fracture

Falling and fractures in elderly men have a great impact on ADL and life prognosis. Moreover, preventing them has become an important issue worldwide. The occurrence of fractures associated with aging is largely due to the decrease in physical function (e.g., muscle mass loss and muscle weakness, frailty, and sarcopenia) in addition to the decrease in BMD [64,65]. Consequently, various clinical conditions caused by testosterone deficiency with age can also affect falls and fractures [2,3,4,5]. Additionally, osteoporosis is a bone condition in which bone mass and strength decrease with aging [8]. Thus, falls are an important factor in the onset of fractures.

Many previous studies investigated the relationship between testosterone and bone fracture risk [45,51,57,58,66,67,68,69,70,71,72,73,74] (Table 3). In addition, several studies have found that elderly men with osteoporotic fractures had statistically significant lower testosterone levels compared with age-matched controls [45,51,66,67,68,70,72,74], and other studies failed to show any associations between testosterone levels and fracture risk [57,58,69,71]. A meta-analysis including 55 studies demonstrated that a significant association was observed in hypogonadism, independent of age, low body mass index, cigarette smoking, excessive alcohol drinking, steroid use, history of diabetes, and so on [62]. Many studies were likely to support the negative effect of testosterone deficiency on the incidences of fall and fracture [45,51,66,67,68,70,72,74], although a smaller number of studies denied this relationship [57,58,69,71]. The more predominant roles for testosterone bone fracture in comparison with BMD may be due to the other role of testosterone in muscle strength and physical performance in men, which leads to sarcopenia and falls risks.

Furthermore, one previous review stated that E2 levels predict the risk of bone fracture, independent of not only serum testosterone and androgens but also estrogens, which are important regulators of bone health in men [75].

## 6. The Effects of TRT on Bone Health among Hypogonadal Men

TRT for hypogonadal men can improve various symptoms (e.g., metabolic syndrome) and has been used worldwide for managing these symptoms and maintaining QOL [2,5]. Moreover, testosterone plays some potential roles in maintaining BMD among men, and TRT is expected to be useful for preventing and managing osteoporosis and improving BMD among hypogonadal men.

Many recent previous studies suggested the ameliorative effect of TRT on osteoporosis/osteopenia [76,77,78,79,80,81,82,83,84] (Table 4). A meta-analysis involving 1083 subjects from 29 randomized controlled studies (RCTs) demonstrated that TRT could improve BMD at the lumbar spine by +3.7% (confidence interval, 1.0%–6.4%) compared with placebo [20]. In particular, many of the previous reports published since 2010 suggest some benefits of TRT on BMD. By contrast, a smaller number of studies failed to find the positive effect of BMD [85,86,87]. However, the subjects of this study were limited to men with hypogonadism, but not always with a lower BMD at baseline. Three studies targeting hypogonadal men with osteopenia or osteoporosis demonstrated that TRT could significantly increase their BMD [76,78,83]. Currently, extremely limited studies, including long-term TRT over 3 years, are available. Further larger and longer RCTs are likely to be required to reach a conclusive result regarding the effects of TRT on bone health and to investigate the benefit of the severity of hypogonadism, degree of baseline BMD loss, and dose of testosterone supplement.

Currently, some guidelines have recommended TRT for symptomatic hypogonadal men with osteoporosis to prevent bone loss and help in acquiring peak bone mass [41,88,89,90,91]. However, the effects of TRT for decreasing the risk of fracture in hypogonadal men with osteopenia and osteoporosis remain unclear. Testosterone therapies have various inverse effects, including erythrocytosis, worsening of prostate hypertrophy and lower urinary tract symptoms, worsening of sleep apnea syndrome, and cardiovascular effects, which are likely to outweigh the beneficial effect for BMD improvement [2,3,5]. Therefore, TRT is not recommended as a tool solely to enhance and maintain BMD for hypogonadal men. At present, vitamin D formulation and antiresorptive drugs are understandably recommended to treat hypogonadism-related osteoporosis in elderly men [92,93,94,95]. On the other hand, TRT should be considered for the patients with any hypogonadal symptoms, and TRT may be an alternative tool for improving hypogonadal symptoms and BMD simultaneously in such cases.

## 7. Conclusions

Testosterone plays an important role in maintaining BMD and bone health among men. In addition, many molecular mechanisms of testosterone on bone metabolism have been currently established by many experimental data. Several recent studies demonstrated the benefit of TRT on BMD, especially in hypogonadal men with osteopenia and osteoporosis. However, a few studies failed to demonstrate its effects, and no evidence supporting the hypothesis that TRT can prevent bone fracture incidence exists. Further studies involving a large number of subjects and longer treatment duration are required to reach a more conclusive result regarding the effects of TRT on bone health. 

Current evidence suggests that TRT is not recommended as a tool solely to enhance and maintain BMD for hypogonadal men. TRT should be considered as one of the treatment options to improve hypogonadal symptoms and BMD simultaneously in symptomatic hypogonadal men with osteopenia.

## Figures and Tables

**Figure 1 jcm-10-00530-f001:**
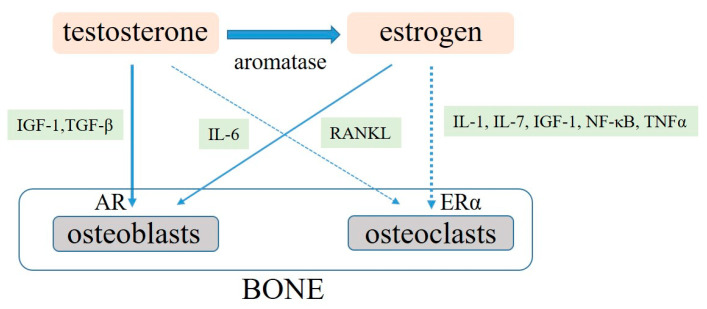
Molecular rules of testosterone and estrogen in bone metabolism. AR, androgen receptor; ERα, estrogen receptor α; IGF-1, insulin growth factor-1; TGFβ, transforming growth factor-β; IL, interleukins; RANKL, receptor activator of NF-κB ligand; NF-κB, nuclear factor-κB; TNFα, tumor necrotic factor-α.

**Table 1 jcm-10-00530-t001:** Prevalence of hypogonadism in men with osteopenia or bone fracture.

Author	Year	Subjects	Prevalence of HG	Reference
Stanley	1991	17 with MTHF	58%	[42]
		61 controls	18%	
Baillie	1992	70 with vertebral fracture	16%	[43]
Kelepouris	1995	47 with atraumatic fracture	15%	[44]
Fink HA	2006	2447 community-dwelling men including 130 with osteoporosis	7%	[45]
Ryan	2011	234 with osteoporosis	24%	[46]
Kotwal N	2018	200 male attendants of patients attending endocrine outpatient department	35%	[47]

HG, hypogonadism; MTHF, minimal trauma hip fracture.

**Table 2 jcm-10-00530-t002:** The relationship between testosterone and BMD in men.

Author	Year	Country	Study Subjects	Hormones	Results	Ref
Greendale (Rancho Bernardo Study)	1997	USA	457 women and 534 men (50–89 years)	TT, DHT, E2, E1, BioT	Higher bioavailable (but not total) testosterone levels were associated with higher BMD.	[49]
Amin (Framingham Study)	2000	USA	448 men (68–96 years)	TT, E2, LH	Hypogonadism related to aging has little influence on BMD.	[50]
Fink	2006	USA	2447 community-dwelling men (>65 years)	TT, E2	The incidence rates of hip bone loss in men with deficient and normal total testosterone were 22.5% and 8.6%.	[45]
Mellström (MrOS Sweden)	2006	Sweden	2908 men (69–80 years)	TT, E2, FT, FE2, SHBG	FT levels were positively correlated with BMD in the hip, femur, and arm but not in the lumbar spine.	[51]
Bjørnerem (Tromsø Study)	2007	Norway	927 women (37–80 years), 894 men (25–80 years)	TT, E2, cFT SHBG	The relationship between all gender steroids and bone loss was weak.	[52]
Kuchuk NO (Longitudinal Ageing Study Amsterdam)	2007	Netherlands	623 men and 634 women (65–88 years)	TT, E2 SHBG	TT had no correlations with BMD	[53]
LeBlanc (MrOS Research)	2009	USA	5995 community-dwelling men (>65 years)	BioT, BioE2 SHBG	The combination of low BioE2, low BaT, and high SHBG was associated with significantly faster rates of BMD loss.	[54]
Vanderschueren (EMAS study)	2010	Belgium	3140 men (40–79 years)	TT, FT, E2 SHBG	TT and FT had no correlations with BMD.	[55]
Cauley (MrOS Research)	2010	USA	1238 men (cross-sectional) 969 men (longitudinal) >65 years	BioT, BioE2 SHBG	No association existed between BioT and hip BMD loss.	[56]
Woo	2012	Hong Kong	1448 men (>65 years)	TT, FT, E2, BioE2, SHBG	TT and FT were not correlated with bone loss.	[57]
Hsu (CHAMP Cohort)	2016	Australia	1705, 1367, and 958 men (>70 men)	TT, DHT, E2 E1, SHBG, LH FSH, cFT	Both TT and cFT had no correlations with bone loss.	[58]

BMD, bone mineral density; TT, total testosterone; DHT, dihydrotestosterone; E2, estradiol; E1, estriol; Bio, bioavailability; LH, luteinizing hormone; FT, free testosterone; FE2, free estradiol; SHBG, sex hormone-binding globulin; cFT, calculated free testosterone; FSH, follicle-stimulating hormone.

**Table 3 jcm-10-00530-t003:** The relationship between testosterone and bone fractures risk.

Author	Year	Country	Study Subjects	Hormones	Results	Ref
Szulc (MINOS study)	2003	France	1040 elderly men (19–85 years)	TT, FT	Hypogonadal men had increased rates of falls and markers of bone resorption.	[66]
Kenny	2005	USA	83 community-dwelling white men (>65 years)	BioT	Fifty-two percent of men with low BioT levels had lower BMD and were likely at an increased risk of fracture.	[67]
Fink	2006	USA	2447 community-dwelling men (>65 years)	TT, E2	Prevalence rates of osteoporosis in men with deficient and normal total testosterone were 12.3% and 6.0%.	[45]
Mellström (MrOS Sweden)	2006	Sweden	2908 men (69–80 years)	TT, E2, FT, FE2, SHBG	FT levels below the median were independent predictors of prevalent osteoporosis-related fractures and X-ray-verified vertebral fractures.	[51]
Tuck	2008	UK	57 men with symptomatic vertebral fractures 57 age-matched controls	FT, BioT, SHBG	Calculated FT was lower in the fracture group than the controls.	[68]
Mellstrom (MrOS Sweden)	2008	Sweden	2639 men (69–80 years)	TT, E2, FT, FE2, SHBG	TT and FT were not significantly associated with fracture risk.	[69]
Meier (The Dubbo Study)	2008	Switzerland	609 men (>60 years)	TT, E2, SHBG	Lower testosterone increased the risk of osteoporotic fracture, particularly with hip and nonvertebral fractures.	[70]
Roddam (EPIC-Oxford Study)	2009	UK	155 men and 281 women	TT, E2, SHBG	There were no associations between fracture risk and testosterone levels.	[71]
Risto	2012	Sweden	39 treated for fracture 45 controls	TT, BioT, BioE2	BioT was a possible marker for increased fracture risk.	[72]
Woo	2012	Hong Kong	1448 men (>65 years)	TT, FT, E2, BioE2, SHBG	TT and FT had no correlations with an increased bone fracture.	[57]
Torremadé-Barreda	2013	Spain	54 men with hip fracture 54 age-matched controls	TT, FT, BioT	Men with hip fractures had significantly lower calculated FT and BiaT levels.	[73]
Hsu (CHAMP Cohort)	2016	Australia	1705, 1367, and 958 men over 70 men	TT, DHT, E2, E1, SHBG, LH FSH, cFT	Both TT and cFT had no correlations with incident fractures.	[58]
Tran	2017	Australia	602 men with incident fractures	TT	TT was significantly correlated with the incidence of fracture risk.	[74]

TT, total testosterone; FT, free testosterone; BioT, bioavailable testosterone; E2, estradiol; FE2, free estradiol; SHBG, sex hormone-binding globulin; BioE2, bioavailable estradiol; DHT, dihydrotestosterone; E1, estriol; LH, luteinizing hormone; FSH, follicle-stimulating hormone; cFT, calculated free testosterone.

**Table 4 jcm-10-00530-t004:** The effects of TRT on BMD in hypogonadal men (from papers published since 2010).

Author	Country	Design	Study Subjects	TRT	Periods	Results	Ref
Kenny (2010)	USA	RCT	131 men with hypogonadism, bone fracture, low BMD, frailty	Transdermal testosterone (5 mg/day)	12–24 months	TRT could increase axial BMD.	[76]
Permpongkosol (2010)	Thailand	Single arm	161 hypogonadal men	1000 mg TU (Nebido)	54–150 weeks	No change in BMD was observed in TRT.	[85]
Aversa (2012)	Italy	Case–control	40 hypogonadal men 20 aged-match control	Intramuscular TU (4 times/year)	36 months	TRT increased vertebral and femoral BMD.	[77]
Wang (2013)	China	RCT	186 men with osteoporosis and hypogonadism	TU (20 or 40 mg/day)	24 months	TRT improved the lumbar spine and femoral neck BMD.	[78]
Bouloux (2013)	UK	RCT	322 men with LOH syndrome	Oral TU (80, 160, 240 mg/day)	1 year	Treatment with oral TU led to BMD improvement.	[79]
Rodriguez-Tolrà (2013)	Spain	Single arm	50 men with LOH syndrome	TG (50 mg/day for 12 months) 1000 mg TU (every 2–3 months from 12–24 months)	2 years	TRT improved lumbar spine and hip BMD.	[80]
Permpongkosol (2016)	Thailand	Single arm	120 hypogonadal men	1000 mg TU (Nebido)	5–8 years	A statistically significant increase was found in vertebral and femoral BMD.	[81]
Rogol (2016)	USA	RCT	306 hypogonadal men	TG (22 or 33 mg)	90–360 days	BMD improved from baseline by TRT.	[82]
Konaka (2016)	Japan	RCT	334 hypogonadal men	TE (250 mg/4 W)	12 months	12-month TRT could not improve BMD.	[86]
Shigehara (2017)	Japan	RCT	74 hypogonadal men with osteopenia	TE (250 mg/4 W)	12 months	TRT for 12 months could improve BMD.	[83]
Snyder (2017)	USA	RCT	211 hypogonadal men	TG (5 mg/day initially)	1 year	TRT increased BMD and bone strength, more in trabecula.	[84]
Ng Tang Fui (2018)	Australia	RCT	100 obese men with hypogonadism	TU (1000 mg/0, 6, 26, 36, and 46 weeks)	56 weeks	No significant changes in the lumbar spine and femoral BMD were observed.	[87]

TRT, testosterone replacement therapy; BMD, bone mineral density; RCT, randomized controlled study; TU, testosterone undecanoate; TG, testosterone gel; TE, testosterone enanthate; LOH, late-onset hypogonadism.

## Data Availability

Data sharing not applicable.

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
