# Peer review of "Testosterone and Bone Health in Men: A Narrative Review"

_jcm, 2021, doi:10.3390/jcm10030530_

Round 1

Reviewer 1 Report

This article reviewed the relationship between testosterone and BMD in men and mentioned the benefits of TRT on BMD among hypogonadal men. Though the authors did a good job of analyzing the literature, the results remain inconclusive. With this, I wonder how this review would help the clinicians.

Minor comments:

  1. Title: Add dash/space between “Minireview”
  2. Please cite the relevant literature in the following lines: 58,62, 140-145, 154, 156
  3. Please revise the conclusion with how this review would be helpful to clinicians.

Author Response

Replies for the comments from reviewer 1

Thank you for your kindly comments and suggestions. I have revised my article according to your comments. The revised points are indicated by red color character. Please re-review my revised manuscript. Thank you again for your contributions.

  1. Title: Add dash/space between “Minireview”

(Reply) Thank you for your comments. According to the recommendation from the editor at the beginning of submission, I have deleted this term of ‘Minireview’.

  1. Please cite the relevant literature in the following lines: 58,62, 140-145, 154, 156

(Reply)

Thank you for your comments. I have added some references which were indicated by you. The references added in the revised article are indicated by red colour.

  1. Please revise the conclusion with how this review would be helpful to clinicians.

(Reply)

Thank you for your suggestions. I have revised the conclusion section by red character.

In particular, I emphasized that TRT is not recommended as only a tool to enhance and maintain BMD for hypogonadal men, and TRT should be considered as one of the treatment options to improve hypogonadal symptoms and BMD simultaneously in symptomatic hypogonadal men with osteopenia. (added in line 201-203 and line 212-215 in page 7)

Reviewer 2 Report

After reviewing and analyzing the manuscript, I describe below issues and changes to be made by the authors.

Title. Change the term Minireview to the term Review Was it a systematic review? How was the review process carried out?

The subject of the work is well-argued, with sufficient data and a large number of studies analyzed, but the structure used in the manuscript is not the most correct

There are two sections in the manuscript, Introduction and Conclusions. The manuscript must have different sections, the description of the methodology used, whether it is a systematic review or not, where it is specified what type of studies are to be included in the review.

A Results section to describe the results obtained in the tables described in the manuscript.

A discussion section to analyze the results obtained. All of these data are within the manuscript but are not consistently described.

Check the bibliography. The articles included in the tables of your manuscript are not updated. The most current study is from 2018.

I hope the proposed changes are made to increase the quality of this article.

Author Response

Thank you for your kind comments and suggestions. I have revised my article according to your comments. The revised points are indicated by red colour character. Please re-review my revised manuscript. Thank you again for your contributions.

  1. Title. Change the term Minireview to the term Review. Was it a systematic review? How was the review process carried out?

(Reply)

Thank you for your comment. According to the recommendation from the editor at the beginning of submission, I have deleted this term of ‘Minireview’. This article is not a systematic review. Rather, this is a narrative review to describe the relationship between testosterone and BMD in men and to mention the benefits of TRT on BMD among hypogonadal men.

  1. The subject of the work is well-argued, with sufficient data and a large number of studies analyzed, but the structure used in the manuscript is not the most correct. There are two sections in the manuscript, Introduction and Conclusions. The manuscript must have different sections, the description of the methodology used, whether it is a systematic review or not, where it is specified what type of studies are to be included in the review. A results section to describe the results obtained in the tables described in the manuscript. A discussion section to analyze the results obtained. All of these data are within the manuscript but are not consistently described. Check the bibliography. The articles included in the tables of your manuscript are not updated. The most current study is from 2018.

(Reply)

Thank you for your comments. As mentioned above, this is not a systematic review or meta-analysis, but this is a narrative review to describe the relationship between testosterone and BMD in men. If the article is a systematic review, I also agree with your comments that methods/results/discussions sections are certainly required. The term ‘mini-review’ in the title has given the reviewers a misleading that this is a systematic review. Please re-review our article as one narrative review. Thanks a lot for your contributions and efforts.

  1. The articles included in the tables of your manuscript are not updated. The most current study is from 2018.

(Reply)

In the present study, I researched many of the previous papers to analyze the relationship between testosterone and bone health. However, I could not find the original articles (excluding meta-analysis and/or systematic review) after 2019 by the Medline.

Reviewer 3 Report

The Authors reviewed the role of testosterone in maintaining BMD and bone health among men. They showed available literature data and suggested TRT should be considered as a tool to enhance BMD in symptomatic hypogonadal men with low BMD. However, long-term trials looking at fragility fracture are needed.

The review is well written; however, the Authors may consider mentioning in their discussion that in hypogonadic men, due to enhanced osteoclast activity, antiresorptive drugs have been proposed to treat the related osteoporosis.

(e.g. Morabito N et al. Neridronate prevents bone loss in patients receiving androgen deprivation therapy for prostate cancer. J Bone Miner Res. 2004 Nov;19(11):1766-70. doi: 10.1359/JBMR.040813;

Smith MR et al. Denosumab HALT Prostate Cancer Study Group. Denosumab in men receiving androgen-deprivation therapy for prostate cancer. N Engl J Med. 2009 Aug 20;361(8):745-55. doi: 10.1056/NEJMoa0809003).

Additionally, low testosterone leads to low E2 production via aromatase and this could provoke deleterious effects on BMD, but also on bone quality as assessed in vivo by different diagnostic tools.

(e.g.  Catalano A et al. Trabecular bone score and quantitative ultrasound measurements in the assessment of bone health in breast cancer survivors assuming aromatase inhibitors. J Endocrinol Invest. 2019 Nov;42(11):1337-1343. doi: 10.1007/s40618-019-01063-0.

Zitzmann M et al. Monitoring bone density in hypogonadal men by quantitative phalangeal ultrasound. Bone. 2002 Sep;31(3):422-9. doi: 10.1016/s8756-3282(02)00831-1.)

Author Response

Thank you for your kind comments and suggestions. I have revised my article according to your comments. The revised points are indicated by red colour character. Please re-review my revised manuscript. Thank you again for your contributions.

1. The review is well written; however, the Authors may consider to mention in their discussion that in hypogonadic men, due to enhanced osteoclast activity, antiresorptive drugs have been proposed to treat the related osteoporosis.(e.g. Morabito N et al. Neridronate prevents bone loss in patients receiving androgen deprivation therapy for prostate cancer. J Bone Miner Res. 2004 Nov;19(11):1766-70. doi: 10.1359/JBMR.040813; Smith MR et al. Denosumab HALT Prostate Cancer Study Group. Denosumab in men receiving androgen-deprivation therapy for prostate cancer. N Engl J Med. 2009 Aug 20;361(8):745-55. doi: 10.1056/NEJMoa0809003).

(Reply)

Thank you for your suggestion. I have added the comments as following; ‘At present, vitamin D formulation and antiresorptive drugs are understandably recommended to treat the hypogonadism-related osteoporosis in elderly men.’ in line 199 to 201 in page 7. In addition, some references which were indicated by you were also added.

2. Additionally, low testosterone leads to low E2 production via aromatase and this could provoke deleterious effects on BMD, but also on bone quality as assessed in vivo by different diagnostic tools. (e.g. Catalano A et al. Trabecular bone score and quantitative ultrasound measurements in the assessment of bone health in breast cancer survivors assuming aromatase inhibitors. J Endocrinol Invest. 2019 Nov;42(11):1337-1343. doi: 10.1007/s40618-019-01063-0. Zitzmann M et al. Monitoring bone density in hypogonadal men by quantitative phalangeal ultrasound. Bone. 2002 Sep;31(3):422-9. doi: 10.1016/s8756-3282(02)00831-1.)

(Reply)

Thank you for your kind comments. I also think that your comment is very important, and agree with your suggestion. I have added the comments as following; ‘Eighty-five 85% of serum E2 derives from testosterone conversion by aromatase in men. Therefore, low testosterone leads to low E2 production via aromatase, which could provoke deleterious effects on BMD, but also on bone quality as assessed in vivo by some different diagnostic tools. A previous study demonstrated that 12-month administration of aromatase inhibitor in elderly men with hypogonadism resulted in a further decrease in BMD.’ in line 138 to 141 on page 4. In addition, some references which were indicated by you were also added.

Round 2

Reviewer 1 Report

Comments addressed 

Author Response

Thank you for your kind comments and suggestions.

I have revised my article according to the comments from reviewer 2. The revised points are indicated by red color character.

Please re-review my revised manuscript.

Thank you again for your contributions.

Reviewer 2 Report

Thanks for your answers and clarifications.

Thank you for specifying that it is a narrative review.
This type of study does not have the same methodology as a systematic review, but it does have a protocol to follow and a specific methodology in the article. (You can look for the Narrative Review Checklist to improve successive publications) To meet part of the requirements it is necessary to add the term "Narrative Review" or the Abstract to the Title. Having searched only one database (Medline) is a limitation of the study. You must add the limitation in the text. Even a narrative review must meet a minimum methodological quality, therefore you must add a paragraph summarizing how the search was and what was searched in that search. The study carried out is very good, it can contribute a lot if the methodological quality is now increased with the suggestions and changes that I suggest.

Author Response

Thank you for your kind comments and suggestions.

I have revised my article according to your comments. The revised points are indicated by red colour character. Please re-review my revised manuscript. Thank you again for your contributions.

(Questions)

Thank you for specifying that it is a narrative review. This type of study does not have the same methodology as a systematic review, but it does have a protocol to follow and a specific methodology in the article. (You can look for the Narrative Review Checklist to improve successive publications) To meet part of the requirements it is necessary to add the term "Narrative Review" or the Abstract to the Title. Having searched only one database (Medline) is a limitation of the study. You must add the limitation in the text. Even a narrative review must meet a minimum methodological quality, therefore you must add a paragraph summarizing how the search was and what was searched in that search. The study carried out is very good, it can contribute a lot if the methodological quality is now increased with the suggestions and changes that I suggest.

(Reply)

Thank you for your suggestions.

I have added the term "A Narrative Review" to the title. In addition, according to the comments, I have added a methods’ section as following; ‘A review of PubMed, MEDLINE, and EMBASE databases were conducted to search for original articles, systematic review, and meta-analysis under key words as following; “testosterone” or “hypogonadism”, “bone mineral density” or “osteoporosis” or “bone fracture”, and “men”. There was no limitation on language, publication status, and study design. Papers published from January 1990 through to October 2020 were collected. We also checked the references of systematic reviews and meta-analyses carefully to identify additional original articles for inclusion. Two reviewers screened the search results, and the data were collected on 4 November 2020. All paper suitable for 3 topics including “The Relationship between Hypogonadism and BMD in human”, “The Relationship between Testosterone and Bone Fracture”, and ‘The Effects of TRT on Bone Health among Hypogonadal Men” from the journal databases were adopted for the present analysis.